# Monitoring the Emergence of Algal Toxins in Shellfish: First Report on Detection of Brevetoxins in French Mediterranean Mussels

**DOI:** 10.3390/md19070393

**Published:** 2021-07-14

**Authors:** Zouher Amzil, Amélie Derrien, Aouregan Terre Terrillon, Audrey Duval, Coralie Connes, Françoise Marco-Miralles, Elisabeth Nézan, Kenneth Neil Mertens

**Affiliations:** 1Ifremer (French Research Institute for Exploitation of the Sea), F-44311 Nantes, France; 2Ifremer, LITTORAL, F-29900 Concarneau, France; amelie.derrien@ifremer.fr (A.D.); aouregan.terre.terrillon@ifremer.fr (A.T.T.); audrey.duval@ifremer.fr (A.D.); kenneth.mertens@ifremer.fr (K.N.M.); 3Ifremer, Station de Corse, F-20600 Bastia, France; coralie.connes@ifremer.fr; 4Ifremer-Centre Méditerranée, F-83507 La Seyne-sur-Mer, France; Francoise.Marco.Miralles@ifremer.fr; 5National Museum of Natural History, DGD-REVE, Station de Biologie Marine de Concarneau, F-29900 Concarneau, France; e.nezan@orange.fr

**Keywords:** lipophilic toxins, LC-MS/MS, brevetoxins, molluscs, *Karenia*, BTXs producer

## Abstract

In France, four groups of lipophilic toxins are currently regulated: okadaic acid/dinophysistoxins, pectenotoxins, yessotoxins and azaspiracids. However, many other families of toxins exist, which can be emerging toxins. Emerging toxins include both toxins recently detected in a specific area of France but not regulated yet (e.g., cyclic imines, ovatoxins) or toxins only detected outside of France (e.g., brevetoxins). To anticipate the introduction to France of these emerging toxins, a monitoring program called EMERGTOX was set up along the French coasts in 2018. The single-laboratory validation of this approach was performed according to the NF V03-110 guidelines by building an accuracy profile. Our specific, reliable and sensitive approach allowed us to detect brevetoxins (BTX-2 and/or BTX-3) in addition to the lipophilic toxins already regulated in France. Brevetoxins were detected for the first time in French Mediterranean mussels (Diana Lagoon, Corsica) in autumn 2018, and regularly every year since during the same seasons (autumn, winter). The maximum content found was 345 µg (BTX-2 + BTX-3)/kg in mussel digestive glands in November 2020. None were detected in oysters sampled at the same site. In addition, a retroactive analysis of preserved mussels demonstrated the presence of BTX-3 in mussels from the same site sampled in November 2015. The detection of BTX could be related to the presence in situ at the same period of four *Karenia* species and two raphidophytes, which all could be potential producers of these toxins. Further investigations are necessary to understand the origin of these toxins.

## 1. Introduction

Brevetoxins (BTXs) are lipid-soluble and heat-stable cyclic polyether compounds first known to be produced by the dinoflagellate *Karenia brevis* (formerly *Gymnodinium breve* and *Ptychodiscus brevis*) [1]. Cultures of *K. brevis* producing BTXs enabled to document a wide variety of these compounds. There are two different known backbone structures: type A BTXs and type B BTXs (Figure 1). The molecular formula of BTX-1 (type A) is C_49_H_70_O_13_, and its structure is made up of 10 rings [2,3]. The molecular formula of BTX-2 is C_50_H_70_O_14_ and its structure is composed of 11 rings [4,5]. Both structures have a lactone function which is considered necessary for their biological activity [6]. BTX-2 is the main toxin isolated from *K. brevis* [6].

The structural elucidation of BTX-2 facilitated the discovery of related chemical structures discovered later. Each family is made up of several compounds that are differentiated in the J-ring in type A BTXs (BTX-1, BTX-7, BTX-10) and in the K-ring in type B BTXs (BTX-2, BTX -3, BTX-5, BTX-6, BTX-8, BTX-9) [7,8] (Figure 1). 

BTX absorption in humans and animals occurs primarily through inhalation and ingestion. Another possible route of exposure exists via skin contact with water contaminated with *K. brevis*, although direct contact with the toxin in water is not well studied. The cell lysis of *K. brevis* releases BTXs into the surrounding water, causing skin irritation [9,10]. BTXs can be transferred from water to air via aerosols, which may cause respiratory tract irritation in humans [7,11] including coughing, sneezing, rhinorrhea, or a burning sensation in the nose and throat [12,13,14]. No human fatalities by inhalation poisoning related to BTXs have ever been reported.

BTXs have been demonstrated in many species of shellfish including mussels, oysters, clams and scallops, which are the main vectors of contamination in humans [15,16]. The consumption of shellfish contaminated with BTXs may result in neurological intoxication in consumers, known as Neurotoxic Shellfish Poisoning (NSP). The symptoms, which appear between one to 24 h after consumption, can include gastrointestinal symptoms, paresthesia, ataxia, bradycardia, dizziness, loss of coordination and coma in more severe cases [17,18]. So far, no fatalities by ingestion poisoning have ever been reported. In addition to effects on human health, blooms of *K. brevis* and other species of the genus *Karenia* have caused recurrent mass mortalities of invertebrates, fish, seabirds, turtles, marine and terrestrial mammals [19,20,21]. 

The species *K. brevis* is recorded in Florida, the Gulf of Mexico, West Indies, and Oceania (New Zealand) [22]. In the Atlantic, it occurs from Mexico to Florida but, in particular, is common on the west coast of Florida [18]. The largest episodes of NSP were recorded in New Zealand between 1992 and 1993 causing illness in more than 180 people following consumption of cockles, greenshell mussels, and oysters [23]. An outbreak of NSP has also been reported in the United States in 1987 (North Carolina) [24] causing illness of 48 individuals who consumed oysters contaminated with BTXs. However, the presence of BTXs has not been reported in Europe to date and therefore, there is no regulatory limit for BTXs in the EU. However, there are several countries in the world where BTXs have been found such as the United States (East Coast, Gulf of Mexico) and New Zealand where regulatory limits have been drawn up. These countries agreed that an acceptable level of exposure to BTX is 20 mouse units/100 g, or 0.8 mg BTX-2 equivalents/kg of total shellfish flesh [16].

In Europe, four groups of lipophilic toxins are currently subject to regulation: okadaic acid (OA)/dinophysistoxins (DTXs), pectenotoxins (PTXs), yessotoxins (YTXs) and azaspiracids (AZAs). In addition to these groups, there are numerous other groups of “emerging” lipophilic toxins that have been shown to be toxic but have only been detected outside of France and Europe (e.g., BTXs, pinnatoxins, palytoxins, ovatoxins). They are not yet regulated due to a lack of sufficient toxicological and epidemiological data needed to establish health safety thresholds. Species producing such lipophilic toxins could be introduced into France via ballast water, biofouling on ship hulls or commercial shellfish trade between countries, and species could have extended their ranges due to climate change.

This caused a need to establish a system for monitoring emerging toxins in shellfish (EMERGTOX), in addition to the French monitoring program for phycotoxins in marine organisms network (REPHYTOX) dedicated to regulated toxins on French coasts. IFREMER launched EMERGTOX in 2018, with the following objectives: (i) the identification of possible hazards for consumers related to the presence of known and currently unregulated toxins in shellfish in France although regulated elsewhere in Europe; (ii) the acquisition of spatio-temporal data series on the main groups of bioaccumulating lipophilic toxins in shellfish identified internationally.

EMERGTOX systematically monitors these emerging lipophilic toxins in shellfish every month at eleven locations along the French coasts, all year round, including phytoplankton toxic alert periods triggered by the REPHYTOX network. In a first step, the LC-MS/MS chemical analysis method used to identify and quantify the target lipophilic toxins was optimized and validated internally in order to screen for regulated lipophilic multi-toxins (OA, DTXs, AZAs, PTXs, YTXs) as well as unregulated toxins (spirolides “SPXs”, gymnodimines “GYMs”, pinnatoxins “PnTXs”, ovatoxins/palytoxins “OVTXs/PLTXs”, brevetoxins “BTXs”). However, this study will only document the first detection of BTXs in France and the phytoplankton species potentially associated with these toxins. Although this first detection was first mentioned in an opinion by the French Agency for Food, Environmental and Occupational Health & Safety (ANSES) [25], this opinion does not provide any scientific details, which are provided here.

## 2. Results

### 2.1. Criteria for Selecting the Shellfish Sampling Points of the EMERGTOX Network

Since January 2018, the LC-MS/MS method for lipophilic multi-toxin chemical analysis has been applied to shellfish samples (mussels, oysters and clams) collected monthly at eleven points along the French coasts (English Channel, Atlantic and Mediterranean). The eleven shellfish sampling points (Figure 2) were selected based on a risk analysis (carried out in conjunction with the risk assessment department at ANSES. The criteria for choosing these sites were as follows:their location in shellfish harvesting areas (mussels, oysters or clams) that are active all year round;an equal geographical distribution over the French coastlines;the existence of historical data for the sampling points, with the aim to obtain long time series;the existence of previous mousse bioassays (which was the reference test prior to 2010 for the monitoring of lipophilic toxins) that remain unexplained (short survival time and/or neurological symptoms);their location outside of risk areas.

**Figure 2 marinedrugs-19-00393-f002:**
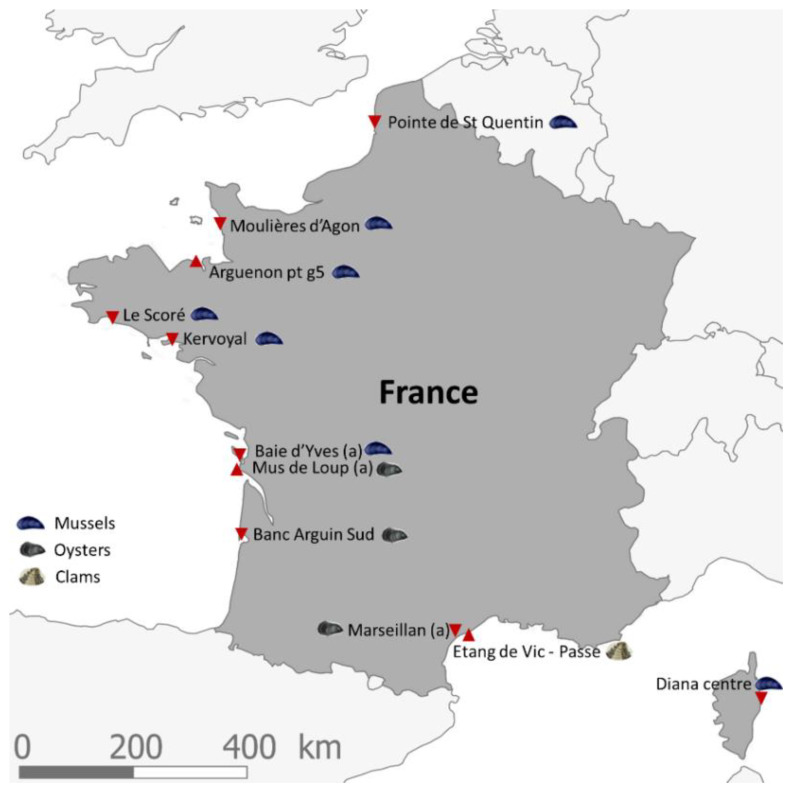
Map of the eleven shellfish sampling points in France within the framework of EMERGTOX designated by filled, red triangles. The respective shellfish that were studied at these localities are indicated by a symbol explained in the legend.

### 2.2. LC-MS/MS Lipophilic Multi-Toxin Quantification in Shellfish

#### 2.2.1. Optimization and Internal Validation of the LC-MS/MS Lipophilic Multi-Toxin Analysis Method in Shellfish

In order to detect the emerging toxins, a new analytical approach was developed to quantify 43 lipophilic toxins in digestive glands (DG) of mussels and oysters. This approach required the use of three LC-MS/MS methods (i.e., three elution gradients and two ionization modes). The single-laboratory validation of this approach was carried out according to the NF V03-110 guideline through construction of an accuracy profile. Excellent performance was obtained with acceptance limits between 15–25% (data not shown) [26].

The EMERGTOX network uses a precise and highly repeatable method. The observed mean recovery rates of BTXs are between 93 and 108%, and the variation in intra-laboratory reproducibility is less than 5%. Such a performance is very satisfactory. It is also important to note that this is a broad-spectrum method, which means that it is possible to test for toxins belonging to several groups in addition to the BTXs (OA/DTXs, YTXs, PTXs, AZAs, SPXs, PnTXs, OVTXs, and microcystins “MCs”). Inevitably, simultaneous testing for such a large number of analogues has a negative impact on method sensitivity. For example, for BTXs, the LOD (limit of detection) is 23 µg/kg DG and the LOQ (limit of quantification) is 70 µg/kg DG. 

All of the results obtained show that, in addition to the detection of various regulated and unregulated groups of lipophilic toxins already listed in France (OA, DTXs, YTXs, PnTXs and SPXs) (results not shown), BTXs were detected for the first time in France in mussels from Diana Lagoon (point Diana-Centre) in Corsica (Mediterranean). Figure 3 compares a LC-MS/MS analysis of the BTX-2 and BTX-3 standards with the same analysis of mussels from Corsica (November 2020).

#### 2.2.2. Detection of BTXs at Diana Lagoon

Table 1 displays all the results on DG of mussels from Corsica tested since autumn 2018. As the analyses were carried out on the DG, the targeted contents (T) were then applied to the total flesh (TF) using the actual percentage (%) of the DG compared with the total flesh (M_DG_/M_TF_ × 100): T_TF_ = T_DG_ × % DG. The extrapolation of the results obtained on the DG to the total flesh assumes that almost all BTXs are concentrated in the DG. Additional analyses were carried out on the remainder of the flesh (excluding the DG) from contaminated mussels sampled in November and December 2018. These analyses did not show the presence of BTXs in the remainder of the mussel flesh.

Except for February 2019 (only detection of BTX-3), more BTX-2 was found than BTX-3 in autumn and/or winter. BTXs levels below the limit of quantification (LOQ; 70 µg/kg DG) were also detected in November 2019; February, March and December 2020; and February, March and April 2021. Figure 4 illustrates the detection and quantification of BTXs in the DG of mussels from Diana Lagoon monitored since January 2018. Since their identification in autumn 2018 as part of EMERGTOX, BTXs have been detected each autumn/winter. Furthermore, an increase in the BTX content is observed to reach a maximum during autumn 2020 (344.6 µg/kg DG, equivalent to 57.2 µg/kg total flesh). Indeed, the concentration of BTXs (BTX-2 + BTX-3) in shellfish was 20 µg/kg in the total flesh in November 2018, 43.8 µg/kg in January 2019 to 57.4 µg/kg in November 2020. In order to show the presence of BTX at levels below the LOQ (70 µg/kg DG, which is still a non-negligible level) on the graph, the maximum value of the LOQ (specified on the legend of Figure 4) was added to these contents. This shows the evolution of these BTXs in autumn/winter between January 2018 and May 2021 (Figure 4). 

Besides mussels, both the DG and remaining flesh of oysters were analyzed as well. These analyses did not show the presence of BTXs either in the DGs or the remaining flesh of oysters. In addition, a retrospective search for these toxins was performed on mussel samples from the “Vigilance” network (predecessor of the EMERGTOX network: prior to 2018, the "Vigilance" network was based on acquiring data on known and emerging or new lipophilic toxins through the use of both mouse bioassays and chemical analyses) for 2015, 2016 and 2017. Samples from the following months were tested: January, February, March, April, June, July, August, September, November and December in 2015; August, September, October, November, December in 2016; and January, February, March, April, June, July, November, December in 2017. None of the targeted BTXs (BTX-2 or BTX-3) were detected in the 2016 or 2017 samples. In contrast, in the November 2015 sample, BTX-3 was measured at a concentration of 135 µg/kg in mussel DGs. 

### 2.3. Identification of Planktonic Species Potentially Producing BTXs in Water Samples from Diana Lagoon

In addition to the monthly mussel samples in Corsica within the EMERGTOX framework, water samples were also obtained from the same site to look for microalgae that are potential BTX producers. Among the dinoflagellates, potential BTX producers that were observed in Diana Lagoon were *Karenia mikimotoi*, *Karenia papilionacea*, *Karenia longicanalis* and an undescribed species *Karenia* sp. 1, all belonging to the family Kareniaceae. Other potential BTX producers were found among the raphidophytes, and the species that were identified in Diana Lagoon were *Fibrocapsa japonica* and *Heterosigma akashiwo* (Figure 5). 

## 3. Discussion

### 3.1. Contamination of Corsica Mussels by BTXs

The results obtained since the EMERGTOX network was set up in January 2018 show that in addition to the lipophilic toxins already listed in France, BTXs (BTX-2 and/or BTX-3) have been detected in France for the first time in mussels from Corsica (Mediterranean), during 2018. Since then, BTXs have been detected regularly at the same period (autumn, winter) every year. The level of BTXs has increased since autumn 2018, with a maximum of 354 µg BTX-2 + BTX-3/kg recorded in the mussels DG during November 2020. 

Currently, as part of the EMERGTOX network, only BTX-2 and BTX-3 are analyzed, among the BTXs, by multi-toxin LC-MS/MS; this covers several groups of regulated and unregulated lipophilic toxins in Europe. Since these BTXs have been detected in France in 2018, the detection should be broadened to include other BTXs. This is why a targeted chemical analysis method specific for BTXs in the total flesh, for which standards are available, should be implemented using LC-MS/MS. Furthermore, it would be useful to develop an analysis method that could be used to detect brevetoxins metabolites that form in shellfish flesh, particularly less lipophilic BTXs. BTXs produced by *K. brevis* can be metabolized/biotransformed in the DG of shellfish [17,27]. These various metabolites have not been found in *K. brevis* cells either in laboratory cultures or during natural blooms [8]. Moreover, in addition to the targeted approach using the LC-MS/MS method that uses the few commercially available BTX standards, a complementary approach needs to take into account the BTXs for which standards are not yet commercially available. An example is the use of the ELISA test [28] to screen for type B BTXs in microalgae and metabolites formed in shellfish flesh.

### 3.2. Phytoplankton Potentially Producing BTXs and Other Metabolites

The dinoflagellates *Karenia brevis* and *Karenia papilionaceae* are confirmed producers of BTXs [29,30]. BTX-like analogues are also produced by raphidophyceae such as *Chattonella marina* [31], *Chattonella marina* var. *antiqua* [29,32], *Fibrocapsa japonica* [31], and *Heterosigma akashiwo* [33].

Although *Karenia brevis* has not yet been recorded on the French coast, greater vigilance is called for because exotic species have already been accidentally introduced in recent years (e.g., via ballast water) [34]. However, it seems far more likely that either one or several of the four recorded *Karenia* species (*K. mikimotoi*, *K. papilionaceae, K.*
*longicanalis* and *K.* sp. 1) and/or the two raphidophytes (*Fibrocapsa japonica* and *Heterosigma akashiwo**)* would be involved in production of BTXs.

Current available data are insufficient to be able to propose a microalgal concentration threshold per liter that can be applied in France (particularly in Corsica) as part of the EMERGTOX network. To compensate for this lack of an alert threshold for potential BTX producers in France, weekly sampling should be adopted in the event of quantification of BTXs in shellfish at a given location. For water analyses, first of all, it is necessary to have a non-fixed raw water sample in order to observe the living cells; this makes it possible to examine the morphological details that can be obscured or modified in Lugol fixed cells. In this study, four *Karenia* species and two raphidophytes have been identified, but possibly other species are present. Detailed morpho-molecular studies need to be carried out to identify the species present and their relation to production of BTXs. Several markers are currently available: the D1–D2 region of LSU 28S rDNA, the 18S SSU (small subunit) rRNA V4 hyper-variable region, and the ITS2 intergenic region (between 5.8S and 28S) are the most frequently used for this group of species [35,36,37].

For *K. papilionacea*, BTX-2 production was confirmed using LC/HRMS and NMR [30]. Other species of microalgae have been suspected of producing BTXs based on ELISA analyses, but the metabolites have not yet been elucidated using physicochemical methods. This is the case for *K. mikimotoi* [38], *K. bicuneiformis* [39], *Chattonella marina*, *C. antiqua* [40], *Heterosigma akashiwo* [34] and *Fibrocapsa japonica* [41].

## 4. Materials and Methods

### 4.1. Materials 

Materials included methanol (LC/MS grade), MilliQ water, ammonium formate (grade: for mass spectrometry), concentrated formic acid 99–100% (HPLC grade), BTX standards (BTX-2, BTX-3; Novakits, France).

Other materials used included glass or plastic bottles, Lugol’s solution, a cooler for transporting the samples, 10 mL sedimentation chambers, inverted light microscope, particle counter (or handheld model). 

### 4.2. Shellfish and Seawater Samples

#### 4.2.1. Seawater for Phytoplankton Observations

For the eleven water points associated with the shellfish points, seawater samples were taken at 1 m water depth using a water sampling bottle (Niskin type) or a pole bottle. The 1-L bottle was filled to 80% of its volume and 3 mL Lugol’s was added to fix the sample. The sample was stored in a cooler until it reached the laboratory where it was observed with an inverted microscope. 

#### 4.2.2. Homogenates of Shellfish for Toxin Analysis

Shellfish samples (mussels, oysters or clams) weighing a total of 2 kg were obtained monthly in sites of the EMERGTOX network, distributed along the French coast (Figure 2). For each raw shellfish sample, the DG were drained, ground and homogenized. When the targeted toxins are present at low concentrations, analysis of the DG produces better detection results because the DG concentrates more trace compounds.

### 4.3. Methods

#### 4.3.1. Microalgae Identification

Upon return to the laboratory, the water sample was left for a few hours to adjust to room temperature. The sample was homogenized by gently stirring the bottle, then 10 mL was sampled using a pipette inserted halfway into the bottle at mid-height. The sample was placed in a sedimentation glass chamber using the Utermöhl method. The chamber was left to settle at room temperature in the dark for eight hours. The phytoplankton species were observed using an inverted microscope at a magnification of ×60 or ×100. Other cells were handpicked from unfixed water sampled using a micropipette and photographed at high resolution using Horiguchi’s method [42].

The targeted phytoplankton taxa were identified using the respective literature [31].

#### 4.3.2. Extraction of Lipophilic Toxins Including BTXs

From the homogenate of 2 kg of harvested shellfish, a subsample of 200 ± 5 mg was placed in a 2-mL Eppendorf tube containing 250 ± 5 mg of glass beads with a diameter of 100–250 µm; subsequently, 945 µL of methanol (MeOH) was added. The sample was ground using Mixer Ball Milling equipment (MM400, Retsch) for 2 min at 30 Hz then centrifuged for 5 min at 15,000× *g* at 4 °C. The supernatant was then transferred to a 2 mL volumetric flask. This operation was repeated, then the two supernatants were combined in the 2 mL flask, and the volume adjusted to 2 mL with MeOH. The extract was transferred to a 2 mL Eppendorf Tube, then 400 μL was filtered through 0.2 μm at 6000× *g* for 1 min at 4 °C before direct analysis by LC-MS/MS [26].

#### 4.3.3. LC-MS/MS Analysis of Toxins

LC-MS/MS analysis was performed on a LC system (UFLC XR, Shimadzu, Marne La Vallee, France) coupled to a hybrid triple quadrupole/linear ion-trap mass spectrometer (API 4000 Qtrap, AB Sciex, Villebon sur Yvette, France) equipped with a heated electrospray ionization (ESI) source. 

Lipophilic toxins were separated on a Kinetex XB-C_18_ (100 × 2.1 mm), 2.6 µm with its pre-column Core-shell, 2.1 mm (Phenomenex) maintained at 40 °C with a 0.3 mL/min flow rate. Mobile phases consisted of water (A) and methanol/water (95:5, *v*/*v*) (B) both containing 2 mM ammonium formate and 50 mM formic acid. The gradient used was: 30–70% B over 1 min, 70–95% B over 9 min, 95–100% B for 0.1 min and maintained 1.9 min, then 100–30% B over 0.1 min, held 3.9 min for equilibration. Injection volume was 5 µL [26].

MS/MS analysis was carried out in positive ion mode with multiple reaction monitoring detection (MRM). The following source settings were used: curtain gas 20 psi, ion spray at 5500 V, turbo gas temperature of 300 °C, gas 1 and 2 set at 40 and 50 psi respectively. Transitions for detection of BTXs and the source settings of mass spectrometer applied on BTXs (BTX-2, BTX-3) were summarized in Table 2. The limit of detection (LOD) of the method was 23 µg/kg GD and limit of quantification (LOQ) was 70 µg/kg GD.

Ionization recoveries: the matrix effects on electrospray ionization of BTXs were estimated by spiking extract of uncontaminated mussels with the BTXs standards. 200 mg of uncontaminated DG of mussels were extracted according to the protocol described in Section 4.3.2. The crude extract was filtered, then 30 µL of doping solution (500 ng/mL of BTX-2 & BTX-3) were added to 270 µL of filtered extract. The final concentration of spiked extract was 50 ng/mL of BTX-2 and BTX-3 mixture. This extract was injected twice during the analysis sequence, before and after samples. The LC-MS/MS results of the field samples analyzed are corrected by ionization recovery.

## 5. Conclusions

BTXs were detected for the first time in 2018 in French shellfish in Corsica (Mediterranean). The maximum reported concentration was 345 µg/kg in the mussel DG for the total of BTX-2 and BTX-3 in November 2020 (corresponding to an estimated value of 57 µg/kg of total flesh). BTXs, a family of lipophilic marine biotoxins, can be produced by several marine microalgae. The dinoflagellates *Karenia papilionacea*, *K. mikimotoi*, *K.*
*longicanalis**,* and *Karenia* sp. 1, and the raphidophytes *Fibrocapsa japonica* and *Heterosigma akashiwo* have been recorded in Diana lagoon, but they have not yet been studied to confirm their production of BTX there. In terms of prospects, it is key to: (i) add an ELISA test to the targeted LC-MS/MS analysis method to broaden the detection to other BTXs for which standards are not available; (ii) isolate, culture and study morphologically and genetically the various potentially BTXs-producing microalgae species observed in situ in Corsica, in order to identify the one or more taxa responsible for the contamination of mussels with BTXs; and (iii) after a data acquisition series, attempt to determine an alert threshold for BTX-producing microalgae, which would trigger BTXs analyses in shellfish.

## Figures and Tables

**Figure 1 marinedrugs-19-00393-f001:**
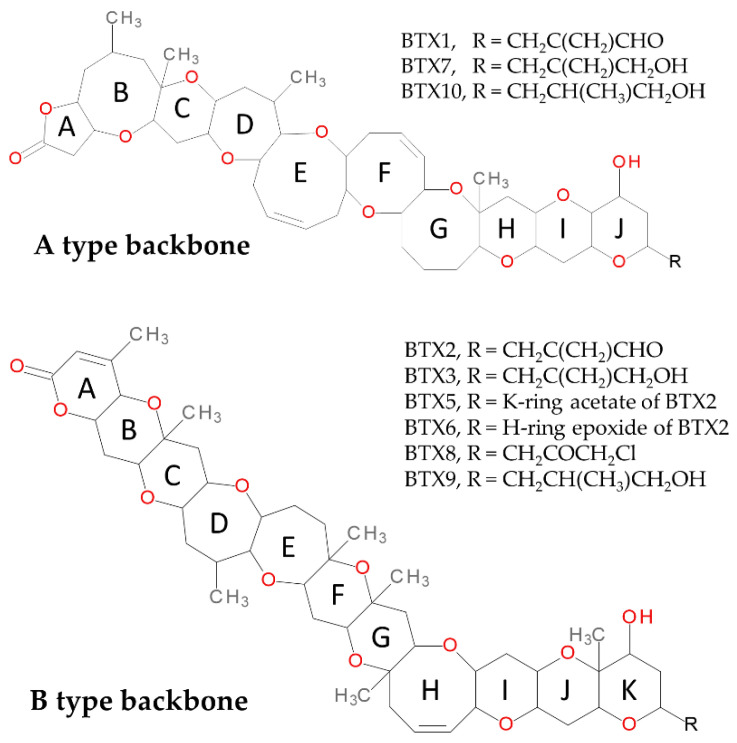
Chemical structures of BTXs identified in environmental samples and cultures of *Karenia*
*brevis*.

**Figure 3 marinedrugs-19-00393-f003:**
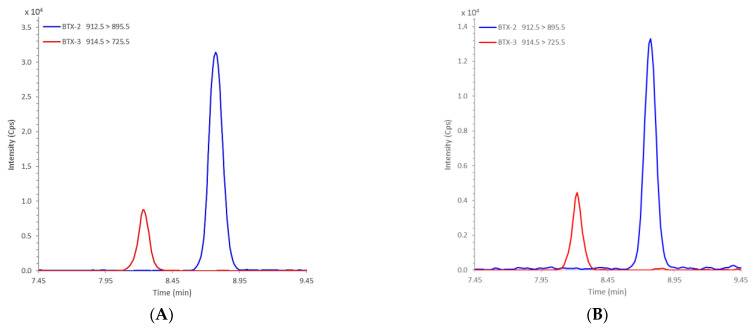
LC-MS/MS analysis of BTX-2 and BTX-3 standards (**A**) and Diana Lagoon mussels from November 2020 containing both BTXs (**B**).

**Figure 4 marinedrugs-19-00393-f004:**
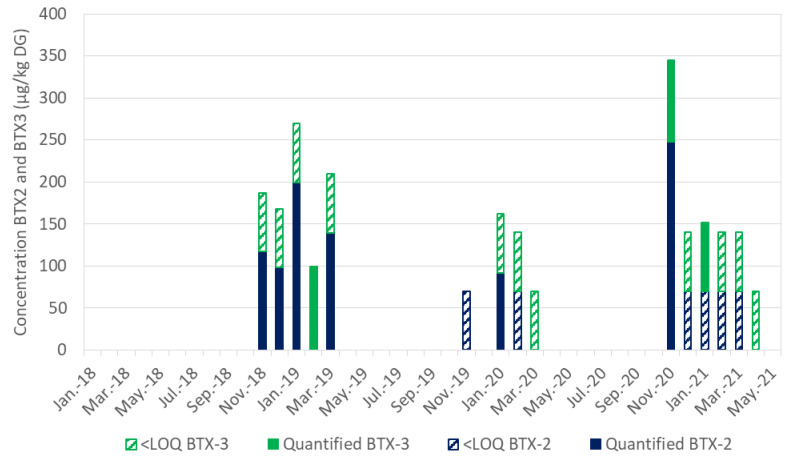
Monthly monitoring of BTXs (BTX-2 and BTX-3) in the digestive glands of mussels in Corsica between January 2018 and May 2021. In order to show the presence of BTXs at levels below the LOQ (70 µg/kg DG, which is still a non-negligible level) on the graph, the maximum value of the LOQ was added to these contents.

**Figure 5 marinedrugs-19-00393-f005:**
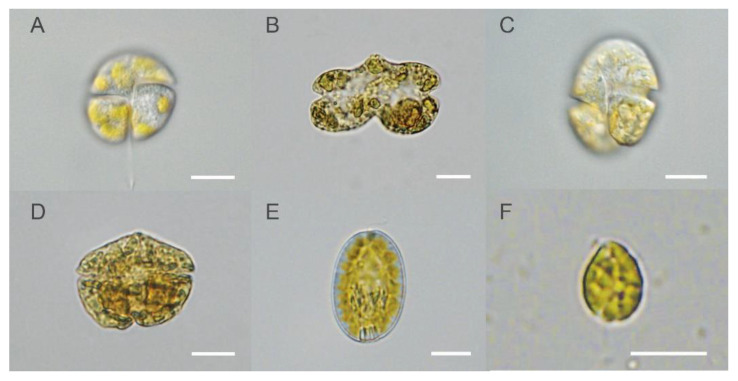
Species observed from Diana Lagoon. (**A**). *Karenia mikimotoi*, living, high focus on ventral area (sampled on 9 December 2019), photo by K.N.M. (**B**). *Karenia papilionaceae,* Lugol’s, mid focus (observed on 7 February 2013), photo by E.N. (**C**). *Karenia longicanalis,* living, high focus on ventral area (sampled on 9 December 2019), photo by K.N.M. (**D**). *Karenia* sp. 1, Lugol’s, mid focus (sampled on 10 December 2018), photo by E.N. (**E**). *Fibrocapsa japonica*, living, mid focus (observed on 24 July 2014), photo by E.N. (**F**). *Heterosigma akashiwo*, living, mid focus (sampled on 17 December 2018), photo by E.N. All scale bars = 10 μm.

**Table 1 marinedrugs-19-00393-t001:** BTX content found in the digestive glands (DG) at Diana Lagoon between January 2018 and May 2021.

Month/Year	Targeted BTX Content (µg/kg DG)		
	BTX-2	BTX-3	BTX-2 + BTX-3
Nov-18	117.2	<LOQ *	117.2
Dec-18	98.2	<LOQ	98.2
Jan-19	199.3	<LOQ	199.3
Feb-19	<LOD **	99.2	99.2
Mar-19	139.5	<LOQ	139.5
Nov-19	<LOQ	<LOD	<LOQ
Jan-20	92	<LOQ	92
Feb-20	<LOQ	<LOQ	<LOQ
Nov-20	247.8	96.8	344.6
Dec-20	<LOQ	<LOQ	<LOQ
Jan-21	<LOQ	82	82
Feb-21	<LOQ	<LOQ	<LOQ
Mar-21	<LOQ	<LOQ	<LOQ
Apr-21	<LOD	<LOQ	<LOQ
May-21	<LOD	<LOD	<LOD

* LOQ (limit of quantification) = 70 µg/kg GD. ** LOD (limit of detection) = 23 µg/kg DG. All the values <LOQ are >LOD.

**Table 2 marinedrugs-19-00393-t002:** Source settings of the used mass spectrometer.

Transitions		ID	Ion	Dwell (ms)	EP (V)	DP (V)	CE (V)	CXP (V)
Q1 Mass (Da)	Q3 Mass (Da)							
912.5	319.3	BTX2	[M + NH_4_]^+^	10	10	81	37	7
912.5	895.5			10	10	81	19	22
914.5	807.5	BTX3	[M + NH_4_]^+^	10	10	84	24	22
914.5	725.5			10	10	84	33	18

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
