# Peer review of "Monitoring the Emergence of Algal Toxins in Shellfish: First Report on Detection of Brevetoxins in French Mediterranean Mussels"

_marinedrugs, 2021, doi:10.3390/md19070393_

Round 1
Reviewer 1 Report
This article reports on the hunt and discovery for brevetoxins in France. This is a significant finding as to the best of my knowledge this is the first report of brevetoxins outside of North America and New Zealand.
The discovery of brevetoxins in Europe is a significant finding and shows the importance of programs like the EMERGTOX. I look forward to seeing follow up articles on analysis of BTX metabolites and finding the BTX producer. FYI - There are some brevetoxin CRMs available from Sigma-Aldrich.
I found the article to clear and detailed enough for the purpose of reporting this finding. Therefore, I recommend that this article is accepted for publication after minor corrections:
L103 & L146 -(AO, DTX...) should be (OA, DTX...)
L166 - LOQ and LOD units are ng/g, I would suggest changing these to µg/kg to be consistent with the rest of the text.
L312 & 316 - no space between the 4 and the degree symbol
L325 & 326 - commas are used instead of decimal points for the times e.g. 0,1 min should be 0.1 min
Font size and or style seems to change several times through the document e.g. L139-L144 or section 3.2
Figure 1 - The resolution is low and pilcrow symbols (¶) shouldn't be present
Reviewer 2 Report
This paper is the first report of BTX occurrence in shellfish from France. The data present BTX analysis in mussels collected over 4 years in the framework of a national monitoring program as well as the identification of cooccurring algal species potentially producing BTX. The authors also provide a brief description of the French monitoring network EMERGTOX as well as the LC-MS/MS multitoxin analysis procedure used in France to screen for lipophilic toxins in shellfish. Although these new findings deserve publication in a journal such as Marine Drugs, major modifications remain needed mostly in the description of the sample preparation, the analytical methodology and the spiking experiment. The paper would also benefit from algal cell counting data as announced in the chapter 2.3. Finally, the authors are highly encouraged to have the manuscript edited for English language.
Specific comments are listed below:
Line 42: add a reference to support the link between lactone group and biological activity
Line 53: exposure via skin contact (“is absorption” may be removed)
Sentences lines 56 to 61 need to be reworded, possibly by removing “In cases of inhalation poisoning,” and “In cases 59 of risk of ingestion poisoning”.
Line 62: scallops are species of molluscs.
Information in Line 59-62 about exposure through food should be part of the following paragraph also about exposure through contaminated seafood.
Line 73: Please provide the number of cases or outbreaks in the New Zealand episode
Line 74 and 75: there is confusion between BTX and BTX poisoning.
The authors should be consistent in the naming of the poisoning (ingestion poisoning, neurotoxic shellfish poisoning, BTX poisoning)
Line 74: BTX singular
Line 80: reference to the 2021 Anses report on BTX should be made
Line 91 REPHYTOX should be in parenthesis
Line 94: is the term risk correct here? Shouldn’t it be a hazard or danger
Line 95: it is unclear which toxins not regulated in France are regulated in Europe. Add this information here.
Line 104 and all along the manuscript: the acronym should be homogenized (with or without an S when in plural)
Line 142: is the list of toxin groups in addition to the BTX? If this is the case, please precise
Line 144: indicate the sensitivity (LOD and LOQ) of the method at least for the BTX group and in Line 163 (precise here what was the LOD), or indicate in the text that those values are provided in the legend of the table. As well, was the recovery of the method assessed?
Line 166: LOD and LOQ are in ng/g, please change to ug/kg to be consistent with the values in the table.
Table 1: <LOQ and <LOD are reported. Are values <LOQ all >LOD? If this is the case, this information should be added to the table legend
Line 172: what statistical analysis was used to state a significant increase in BTX concentrations over time?
Figure 4: This figure is difficult to understand and it is unclear what additional information it provides from table 1. The figure does not stand alone. The legend would benefit from the explanation provided in lines 175-176.
The figure should be deleted and the March 2020 as well as the 2021 data reported in the table.
The other option would be to delete the table and keep the figure as it illustrates well the seasonal variations.
Figure 4 legend: CorScheme and December 2020 should be defined
Line 184-190: could this information be provided in Figure 4?
Line 191: the title needs to be reworded. The last part “and associated shellfish samples in Corsica” does not make sense. In addition, the corresponding chapter does not report abundance.
Line 193-199: abundance data should be provided
Line 229: delete “to”
Line 230: reference for the ELISA should be provided
Line 232-241: the Anses report may be referenced here
Line 248: reference is missing on exotic species introduced in France.
Line 278-279: The source and model of the equipment listed should be provided here.
Change the title of 4.3 to be more specific and not a repeat of Title 4.
Line 296: “left for a while”: please provide a range time in minutes or hours.
Line 300: enumeration of algal cells are not provided in the manuscript
Line 307-316: the spiking experiment should be further detailed by providing information on the concentrations used, the number of replicates, the species used as blank matrix, how and at which step of the process the spike was added etc.
In this paragraph, it is unclear if 200 mg correspond to the matrix used for the spike or that of the field samples analyzed.
Was the recovery of the method assessed?
Line 347: replace accumulation with contamination of mussels with BTX. The algae are indeed not responsible for the accumulation per se.
Round 2
Reviewer 2 Report
All comments have been addressed except the 2 following minor points that remain to be added. The paper will be acceptable for publication in Marine Drugs journal after those minor revisions.
Table 1: although indicated as done in the response to the reviewer, the legend of table 1 still need precisions that values <LOQ are >LOD. This information is important as it shows that although not quantified, BTXs are detected most often the time.
Please add the Anses Opinion on BTX which was published online in June 2021 (English version) https://www.anses.fr/en/system/files/ERCA2020SA0020EN.pdf
